# Population size, habitat association, and local residents' attitude towards rock hyrax (*Procavia capensis*) in Zegie Peninsula, Ethiopia

**Birkie Alehegn, Zewdu Kifle** [ORCID]*

Department of Biology, Bahir Dar University, Bahir Dar, Ethiopia

* zewdu96@yahoo.com

## Abstract

Given the current rate of habitat loss, particularly in the tropics, reliable data on the population size and habitats of wild animals are crucial for initiating conservation and management activities in specific areas. Wildlife ecologists have not studied the ecology of most medium-sized mammals in detail. The rock hyrax (*Procavia capensis*) is one such medium-sized mammal that has not been well-studied in Ethiopia. Therefore, we conducted this study to determine the population size and examine local residents' attitudes toward the rock hyrax in the Zegie Peninsula, Ethiopia. We applied the strip-transect counting method to estimate the population size and determine the density of rock hyrax in the area. We used structured questionnaires and informant interviews to assess local people's attitudes. We counted a total of 469 rock hyrax individuals in the study area. Adults accounted for 72.4% of the population, while juveniles comprised 27.6%. Through extrapolation, we estimated a total population of approximately 2,184 rock hyraxes within the Zegie Peninsula, with a density of 36.5 individuals/km². The juvenile-to-adult ratio was 1:2.6 in the area. Rock hyraxes were most frequently observed in lakeshore habitats, followed by residential areas. Most respondents held negative attitudes toward the species, citing environmental pollution (notably a bad smell) in the church and residential compounds due to hyrax pellets and urine, as well as damage to fruits and vegetables. Respondents also described a traditional belief that rock hyraxes possess magical powers enabling them to steal food when they see people eating. Additionally, most reported that hyrax body parts are used to prepare traditional medicines for human and cattle ailments. Key informants specified their use in treating cattle anthrax and leprosy (via pellets). Most respondents opposed hyrax conservation in the area. These findings provide a baseline for developing conservation management strategies to ensure the species' long-term survival.

**Data availability statement:** All relevant data are within the paper and its Supporting information file.

**Funding:** This work was funded by the Bahir Dar University The funders had no role in study design, data collection and analysis, decision to publish, or preparation of the manuscript.

**Competing interests:** The authors have declared that no competing interests exist.

## Introduction

The biological diversity around the globe represents an irreplaceable resource critical to current and future generations [1]. However, escalating demands to feed a rapidly growing human population have driven widespread deforestation for agriculture, live-stock grazing, settlement expansion, and urbanization [2,3]. These activities fragment landscapes and disrupt the distribution, abundance, and ecological dynamics of wild flora and fauna [4]. The ecological consequences of habitat fragmentation on biodiversity are multifaceted, varying in both magnitude and long-term impacts [4,5]. In Africa, where human populations are expanding at unprecedented rates, the need for land to accommodate housing and crop cultivation has intensified human-wildlife conflicts [6]. Human-driven habitat destruction, particularly through agricultural encroachment, remains a primary driver of biodiversity loss across the continent [6].

Ethiopia's dramatic altitudinal gradients and diverse topography form one of the most ecologically varied landscapes in Africa [7]. Elevation spans from 126 m below sea level in the Danakil Depression (Afar region) to 4,620 m above sea level at Ras Dejen, the highest peak near the Simien Mountains National Park. This vast altitudinal range fosters climatic extremes, ranging from arid lowland deserts to cold, humid alpine zones. Such environmental heterogeneity underpins Ethiopia's exceptional diversity of terrestrial and aquatic ecosystems, which support rich floral and faunal, including high rates of species endemism [8,9].

Hyraxes are medium-sized terrestrial mammals native to Africa and the Middle East. They comprise three extant genera: *Procavia*, *Heterohyrax,* and *Dendrohyrax*. Five extant species of hyraxes are recognized: the rock hyrax (*Procavia capensis*) and the yellow-spotted rock hyrax (*Heterohyrax brucei*), which both inhabit rock outcrops, including cliffs in Ethiopia and isolated granite formations (known as *koppies*) in southern Africa; the western tree hyrax (*Dendrohyrax dorsalis*), southern tree hyrax (*D. arboreus*), and eastern tree hyrax (*D. validus*). Their distribution is restricted to Africa, except for *P. capensis*, which is also found in the Middle East.

Rock hyraxes, particularly *P. capensis,* resemble large guinea pigs or rabbits, with rounded short ears and no visible tail. Their coat color ranges from yellowish to grayish-brown, with a lighter underbelly. Despite their elongated bodies and short legs, their hunched posture and long fur obscure these features [10]. Rock hyraxes also possess long, tactile hairs, especially on the muzzle, throat, cheeks, and rump. Their diet consists primarily of grasses, fruits, and leaves [10,11]. They are characterized by disproportionately large jaws and the ability to retract their lips fully, enabling them to bite off and chew large mouthfuls of vegetation in one motion. Their habitats include savannas, shrublands, deserts, and rocky areas such as cliffs and mountain peaks [10]. Rock hyraxes are slow-moving and have few defensive mechanisms against predators. They rely heavily on rock outcrops, which offer protection from predators, access to safe foraging areas, and elevated vantage points [12].

Rock hyraxes (*P. capensis*) inhabit rock outcrops, boulder piles, and fractured cliff faces [13]. They exhibit the broadest geographical and altitudinal distribution among hyrax species [14]. They play a vital role in arid and rocky ecosystems, serving both

as prey for predators and seed dispersers for certain plant species. Although there are no major threats to their survival, population dynamics are regulated by predation, intraspecific competition, immigration, territorial conflicts, and dispersal. Humans hunt them for their meat and fur, for example, the Hadza (or Watindiga), an indigenous hunter-gatherer group in Tanzania, relies on rock hyraxes as a food source [15]. Similarly, they are hunted for sustenance in parts of Egypt [16]. In regions such as Kenya and Israel, rock hyraxes are excluded from human settlements due to their role as reservoirs of cutaneous *leishmaniasis* [17].

Rock hyraxes are classified as Least Concern on the IUCN Red List of Threatened Species [18]. However, their populations have declined due to habitat loss, human encroachment, disease, and predation [19]. They are considered pests in some regions, damaging farmers' crops and causing agricultural conflicts. Additionally, they face threats from both avian and mammalian predators.

Assessing the population size of wild animals, their habitat associations, and understanding local communities' attitudes toward wildlife are critical for designing sustainable conservation and management strategies that balance ecological integrity with socioeconomic needs [20]. Population size estimates are foundational for assessing a species' conservation status and elucidating habitat associations; for example, the rock hyrax's reliance on rocky outcrops and crevices makes it particularly vulnerable to landscape alterations. To date, few studies have assessed the population size, distribution, and ecology of rock hyraxes in Ethiopia [21–23]. Consequently, the species' population size and distribution remain poorly documented and understood across the country. Specifically, no published data exist on the population size of rock hyraxes in the Zegie Peninsula, Ethiopia. Furthermore, local communities' attitudes toward the species have not been studied in the country. Local residents' attitudes, whether shaped by cultural significance, perceived conflicts, or economic impacts, can profoundly influence the success of conservation initiatives. This study aimed to: (1) estimate the population size of rock hyraxes in the Zegie Peninsula; (2) assess their habitat associations in the area; and (3) examine local people's attitudes toward rock hyrax conservation. The findings provide baseline data for future research including studies on traditional medicinal uses of rock hyraxes and other ecological and behavioral aspects in the region.

## Materials and methods

### Ethical considerations

This study was carried out in adherence to ethical guidelines approved by the Human Research Ethics Review Board of Bahir Dar University (BDU, Ethiopia). Prior to participation, respondents were informed of the study objectives, the nature of the data to be collected, and how their information would be utilized. Investigators assured strict confidentiality in data handling. Verbal consent was obtained from all participants after providing a detailed explanation of the survey's purpose and the interview procedure.

### Study area

We carried out this study in the Zegie Peninsula, located on the southwestern shore of Lake Tana, Ethiopia. The peninsula covers an area of 13.47 km² and lies between 11°40' to 11°43' N latitude and 37°19' to 37°21' E longitude. Its elevation ranges from 1,775–1,985 m above sea level. The Zegie Peninsula is renowned for its Ethiopian Orthodox Tewahedo churches and monasteries, some dating back to the 13th century. These sites hold significant cultural, religious, historical, and aesthetic value, making the area a vital hub for preserving Ethiopia's religious traditions and heritage. It is a prominent tourist destination. The climate is classified as a warm temperate highland monsoon, with a mean annual temperature of 21.9 °C. The region experiences two distinct seasons: a wet season (June—October and a dry season (November—May.

The Zegie Peninsula is bordered by Lake Tana to the north and east, wetlands to the west, and farmlands to the south. Dominant vegetation in the area includes *Mimusops kumme*, *Syzygium guineense, Ficus vasta*, *Jasminum grandiflorum*, *Momordica foetida*, *Clematis simensis*, *Tragia brevipes*, *Urera hypselodendron* and *Phytolacca dodecandra.* The

peninsula also supports diverse mammal species, including the leopard (*Panthera pardus*), grivet monkey (*Chlorocebus aethiops*), rock hyrax (*Procavia capensis*), bushpig (*Potamochoerus larvatus*), and duiker (*Sylvicapra grimmia*). In addition, the peninsula supports diverse species of birds, amphibians, reptiles, mollusks, and insects. However, its wild animals and their habitats are under threat due to deforestation and illegal hunting practices [24]. For example, common bushbuck (*Tragelaphus scriptus*) and warthog (*Phacochoerus africanus*) have been eradicated due to illegal hunting in the area [25].

## Population size and habitat association of rock hyraxes

From January 10 to June 15, 2021, we applied the line transect method to estimate the rock hyrax population size in the study area. The census zone was divided into sampling units using a two-stage sampling technique. We divided the study area into four habitat types: lakeshore coffee-dominated semi-natural forest, human residential areas, tall-tree coffee plantations, and bushland. We established 18 transect lines across these habitats, with lengths proportional to the size of each stratum. Average transect lengths were 4.12 km in human residential areas, 4.42 km in forested zones, 1.83 km in bushland, and 0.69 km in lakeshore habitats. Consecutive transect lines were spaced 200–300 m apart to minimize double counting risk, ensuring negligible likelihood of duplicate observations.

During the survey, we recorded rock hyrax numbers, age categories, and habitat types. We walked transects on three consecutive days per month during active foraging and sun-basking period: mornings (9:00–11:30 a.m.) and late afternoons (2:30–5:00 p.m.), at an average speed of 1 km/hr [26]. A fixed 25 m width (50 m total strip width) was surveyed on either side of the transect line [27]. Every 50 m, we paused for two minutes to scan vegetation and rocky outcrops for resting individuals. Sampling covered 2.71 km2 (21.5%) of the total 12.62 km2 study area, spanning dry (February–April) and wet (June, July, and September) seasons. We repeated transects across seasons and times of day to account for temporal variation.

## Questionnaire survey

Between March 16 and June 20, 2021, we conducted structured questionnaire surveys to assess local residents' attitudes toward rock hyraxes. The questionnaire was pretested with 17 respondents to refine clarity and ensure validity and reliability within the study context. We applied a two-stage sampling technique: first, we purposively selected seven villages from two *kebele* administrative units (Ura and Yiganda-Mehale Zegie); second, we systematically selected household heads (respondents) using official registries from the kebele offices. We interviewed 210 respondents (18.82% sampling proportion) out of 1,116 households inhabiting the peninsula. We excluded seven incomplete questionnaires from analysis.

We designed the structured questionnaire to examine local attitudes, practices, beliefs, and knowledge regarding the ecological and cultural significance of rock hyraxes, as well as to identify factors influencing community perceptions. It also included respondents' education levels, ages, sexes, and livelihoods. To triangulate the questionnaire findings, we conducted key informant interviews with nine individuals (community leaders, elders, religious leaders, and association leaders) to gather insights into cultural practices and traditions associated with rock hyraxes.

## Data analysis

We summed up data from all transects within each habitat to estimate the total rock hyrax population [28]. We extrapolated the transect results to estimate the total population size across the study area. We excluded bushland habitats from the analysis, as no rock hyraxes were recorded there during the study period. We determined population density by dividing the total number of individuals counted in sampled sites by the total land area of the study area (km2). We analyzed data using descriptive statistics (frequencies and percentages). We used the chi-square test of independence to assess associations between categorical variables (gender vs. attitude toward rock hyrax, and education level vs. attitude toward rock hyrax). All tests used a significance level of $\alpha = 0.05$.

## Results

### Population size of rock hyrax

We counted a total of 469 rock hyrax individuals in the study area. Of these, 339.5 individuals (72.4%) were adults and 129.5 (27.6%) were juveniles. Through extrapolation, the total estimated population size was 2,184 individuals (95% confidence interval: 1,720–2,648) across the Zegie Peninsula. The population density was 37.2 individuals/km$^2$, with a juvenile-to-adult ratio of 1:2.6.

From the total population, we counted 426 rock hyrax individuals during the dry season and 512 during the wet season (Table 1). During the dry season, adults comprised 75.1% of the population, while babies (juveniles) accounted for 24.9. In the wet season, adults constituted 70.1% and babies 29.9%. The juvenile-to-adult ratio was 1:3.0 in the dry season and 1:2.3 in the wet season. Population density also varied seasonally, with 33 individuals/km$^2$ recorded in the dry season and 40 individuals/km$^2$ in the wet season.

We recorded the highest numbers of rock hyrax individuals in the lakeshore habitat (223.5 individuals, 47.7% of total observations), followed by the human residential area (210 individuals, 44.8%), and the forest habitat (35.5 individuals, 7.5%) (Table 1). We did not record rock hyrax in the bushland habitat. We recorded the highest population size of rock hyraxes in the lakeshore habitat during both the dry season (210 individuals) and the wet season (237 individuals).

### Questionnaire survey

**Socio-demographic features of the respondents.** We conducted questionnaire-based interviews with 203 respondents, among whom 105 were males and 98 were females with ages ranging from 21 to 65 (Table 2). Most

**Table 1. Population size and structure of the rock hyrax during the dry and wet seasons.**

| Habitat type | Season | | | | | | | |
|---|---|---|---|---|---|---|---|---|
| | Dry | | | | Wet | | | |
| | Adult | Juvenile | Total | % | Adult | Juvenile | Total | % |
| Human residence | 137 | 51 | 188 | 44 | 162 | 70 | 232 | 45 |
| Forest | 26 | 2 | 28 | 7 | 31 | 12 | 43 | 8 |
| Lakeshore | 157 | 53 | 210 | 49 | 166 | 71 | 237 | 46 |
| Bushland | – | – | – | | – | – | – | |
| Total | 320 | 106 | 426 | – | 359 | 153 | 512 | – |

**Table 2. Socioeconomic and demographic profile of the respondents.**

| Characteristics | Frequency | Percentage |
|---|---|---|
| Sex | | |
| Male | 105 | 51.7 |
| Female | 98 | 48.3 |
| Age | | |
| 21-35 | 29 | 14.3 |
| Above 35 | 174 | 85.7 |
| Educational level | | |
| No education | 59 | 29.1 |
| Primary school | 43 | 21.2 |
| High school | 31 | 15.3 |
| Higher education level (above grade 12) | 18 | 8.8 |
| Religion (priests and deacons) | 52 | 25,6 |

(97.5%) respondents reported that their livelihood depended on coffee plantations, fruit production, fishing, selling wood (log) for firewood, small-scale trading, and handcraft making and selling. Most (97.0%) respondents owned farmland. Most (71.4%) respondents did not own sheep, while the remaining 28.6% kept sheep. A few (2.5%) respondents were employed by governmental and non-governmental organizations. Notably, 29.1% of respondents had not attended both formal and informal education.

**Local residents' attitudes towards rock hyraxes.** Over half (57.6%) of respondents held negative attitudes towards rock hyraxes, while the remainder (42.4%) expressed positive views (Table 3). Respondents with positive attitudes cited using rock hyraxes as a food source for domestic dogs and generating income by selling their hides and meat to traditional healers. Male respondents exhibited more favorable attitudes toward the presence of rock hyraxes than female respondents, with a statistically significant relationship between gender and attitudes ($\chi 2 = 27.37$, df = 1, $P = 0.001$). Similarly, respondents with higher educational attainment expressed stronger support for rock hyrax presence. Attitudes toward the species also differed significantly across educational categories ($\chi 2 = 90.91$, df = 4, $P = 0.001$).

Most (64.5%) respondents opposed the conservation of rock hyraxes in the area, while the remaining (35.5%) supported it. Opposition was highest among respondents with no formal and religious education and those with primary-level education; however, many secondary school graduates and individuals with higher education levels predominantly favored conservation (Fig 1). A statistically significant relationship was observed across educational attainment categories toward rock hyrax conservation ($\chi 2 = 82.26$, df = 4, $P = 0.001$).

**Perceived costs and benefits of rock hyrax presence.** Most (80.0%) respondents reported that rock hyraxes damaged fruits and vegetables, while a minority (20.0%) believed that the species did not cause significant crop damage in their localities. Key informants similarly emphasized the destructive behavior of rock hyraxes on crops. Additionally, many respondents noted that economically important plants such as *Rhamnus prinoides*, *Carica papaya*, *Citrus lemon*, *Eucalyptus camaldulesis*, *Mimusops kummel*, and *Acacia pilispina* were damaged by rock hyraxes. Key informants further underscored the animals' destructive impact on plants.

Many local residents held negative traditional beliefs about rock hyraxes, associating them with magical powers (locally termed "*wosaji*"). They claimed that rock hyraxes magically steal food when they see people eating. In addition, another common problem reported by key informants was the occurrence of rock hyrax pellets and urine, which caused environmental pollution (notably a bad smell) in and around churches and residential compounds.

Most (93.0%) respondents stated that rock hyraxes can bring economic benefits to humans. Of these, 88.0% described that rock hyraxes' body parts are used in preparing traditional medicines to treat human and cattle ailments, while 3.0%

**Table 3. Attitude of respondents toward rock hyrax.**

| Variables | Respondents' feeling | |
|---|---|---|
| | **Good** | **Bad** |
| Educational level | | |
| Primary school | 21 (48.8%) | 22 (51.2%) |
| Secondary school | 27 (90.0%) | 3 (10.0%) |
| Higher education level (above grade 12) | 19 (100%) | – |
| Religion (priests and deacons) | 16 (30.8%) | 36 (69.2%) |
| No education | 3 (5.1%) | 56 (94.9%) |
| *Total* | *86 (42.4%)* | *117 (57.6)* |
| Gender | | |
| Male | 51 (48.6%) | 54 (51.4%) |
| Female | 14 (14.3%) | 84 (85.7%) |
| *Total* | 65 (32.0%) | 138 (68.0%) |

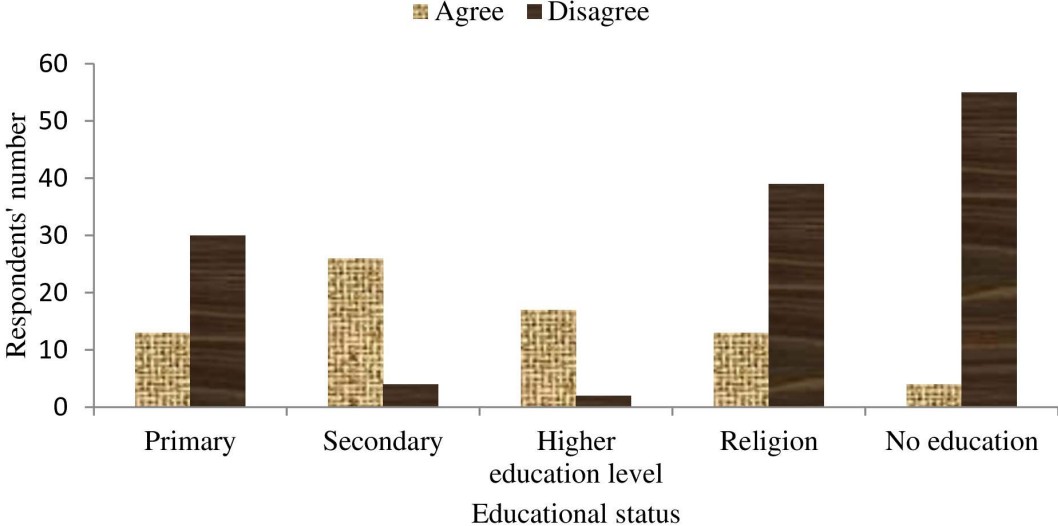

**Fig 1. Proportion of respondents supporting rock hyrax conservation in the region (religion = priests and deacons through informal education); (Higher education level = above grade 12).**

cited the use of their hides (though specific diseases were unspecified). Key informants also highlighted the species' medicinal value, noting that its flesh is used to treat anthrax in cattle and its pellets to address leprosy. Additionally, a small proportion of few respondents (2.0%) mentioned using rock hyraxes as dog food.

**Awareness of rock hyrax population and conservation status.** When we asked about the population size of rock hyraxes, 112 (55.0%) respondents reported a decline in the area, 76 (37.0%) believed the population had increased in recent years, and 15 (8.0%) stated they had no information about population trends. Over half of respondents (56.0%) most frequently observed groups of three rock hyraxes, while the remaining 44.0% reported seeing groups of four to five individuals. Most key informants also mentioned that rock hyraxes are no longer as commonly sighted in lakeshore areas and church compounds as they were in previous years. They reported that hunting for traditional medicine, predation by dogs, and habitat disturbances such as tree cutting and stone fencing construction are responsible for their decrement in the area.

**Controlling mechanisms.** Many (57.0%) respondents employed lethal methods to control crop damage caused by rock hyraxes and prevent pollution from their pellets and urine, while the remaining 43.0% relied on chasing. Local residents killed rock hyraxes using spears, poison, traps or dogs. They chased them away by throwing stones, deploying dogs, and making noise.

**Conservation threats to rock hyraxes.** In the Zegie Peninsula, 115 (56.0%) respondents reported threats to rock hyraxes in their locality, while the remaining 44.0% perceived no threats to the species. Among the 115 respondents who identified threats, most (65.2%) considered hunting and killing as the primary threat followed by predation by dogs (28.8%) and human-induced habitat disturbance (6.0%).

## Discussion

Most researchers have paid relatively little attention to the population and behavioral ecology of medium-sized mammals and their conflicts with local people. In this study, we assessed the population size of rock hyraxes and the extent of human–rock hyrax conflict in the Zegie Peninsula, Ethiopia. We documented 469 rock hyrax individuals in the peninsula, with adults significantly outnumbering babies (juveniles). A similar dominance of adults over babies has been reported

in Bale Mountains National Park, Ethiopia [13]. This could be due to higher predation pressure on young individuals. Additionally, juveniles may remain concealed in shelters more frequently, making them less detectable during systematic transect surveys.

In the present study, the average density of rock hyraxes was 37.2 individuals/km². In contrast, mean population densities of 664.5 and 529.9 individuals/km² during the wet and dry seasons, respectively, were reported in Bale Mountain National Park, Ethiopia [13]. Studies elsewhere on rock hyrax population density have also documented ranges of 20–100 individuals/km² in Mount Kenya. Kenya [29], 500–4,000 individuals/km² in Serengeti National Park, Tanzania [19], and 73–94 individuals/km² in Matobo National Park, Zimbabwe [28]. The limited availability of rock outcrops, cliffs, caves, and gorges with crevices or rock piles in the Zegie Peninsula may be the primary cause of its low hyrax population density. Furthermore, high habitat disturbance in this unprotected study area likely exacerbates the low density of rock hyraxes. Severe habitat alterations caused by human activities in natural ecosystems are known to influence wildlife ecology and population dynamics [30], and such disturbances similarly reduce the population size and density of rock hyraxes. Protected areas like Bale Mountains National Park (Ethiopia) and Serengeti National Park (Tanzania) are expected to support higher densities of wild animals compared to human-modified landscapes such as the Zegie Peninsula.

Extreme population fluctuations of hyraxes have been recorded in Montopos, Zimbabwe [11], and Serengeti National Park, Tanzania. [19]. In Augrabies Falls National Park, South Africa, hyrax populations fluctuate primarily due to predation, food availability, drought, and infectious disease [31]. In the present study, we counted a higher number of rock hyraxes during the wet season than during the dry season. This increase might be attributed to higher birth rates during the wet season, when greater food availability aligns the nutritional demands of the animal. The abundance of nutritious food items in the wet season may also promote breeding. Seasonal fluctuation in hyrax populations are primarily influenced predation and food availability [31,32]. During the dry season, hyraxes face heightened predation risk as they travel farther from their shelters and refugees in search of scare food resources [13,31].

Rock hyraxes were not uniformly distributed across the study area. For example, no individuals were recorded in bushland habitats, as these habitats might lack suitable conditions for foraging and shelter. However, we counted a higher number of rock hyraxes in the lakeshore habitats, likely due to abundance rock outcrops that provide for foraging opportunities and refuge. Similarly, rock hyrax distributions are non-uniform in Bale Mountains National Park, Ethiopia [13]. Their distribution primarily depends on the availability of food and shelter, which enable them to evade and monitor predators effectively. Suitable shelter availability strongly influences rock hyrax distribution patterns [29]. Likewise, rock hyraxes are typically associated with rock outcrops, cliffs, or boulder piles interspersed with bushes [13,33]. Their presence also declines near human settlements due to habitat loss and human disturbance [34].

Interestingly, a significant number of rock hyraxes inhabited human dwellings in the current study area, likely because local residents construct rock fences that offer hiding places from predator. These structures may provide protection, illustrating how hyraxes adapt to human-altered landscapes by exploiting available resources. Additionally, as animals increasingly inhabit human-modified landscapes, they adapt to human disturbances by utilizing available resources. Such ecological and behavioral flexibility may enable them to roam and forage near human dwellings. However, further investigation into habitat characteristics is required to confirm this explanation in the study area.

Most residents in the Zegie Peninsula held negative attitudes toward rock hyraxes in their locality. This likely stems from the species' destructive impacts on fruits and vegetables. Additionally, a traditional belief persists among locals that rock hyraxes possess magical powers to steal food, further contributing to their unfavorable perception. Rock hyraxes are also known reservoirs of cutaneous *leishmaniasis,* posing a risk of disease outbreaks near human settlements [17,35]. Colonies habitually urinate and defecate in communal sites, which can produce unpleasant odors in churches and residential compounds. These behaviors likely reinforce the community's negative views of the species. While traditional medicine practices may indirectly promote conservation by valuing certain wildlife, local communities in this region remain wary of rock hyraxes due to their perceived ecological and cultural impacts.

In the present study, respondents reported that rock hyraxes damages crops. When these animals inhabit boulders or cliffs adjacent farmlands, they damage crop seedlings [13. 21]. Studies have also documented crop damage by rock hyraxes inhabiting near agricultural areas [36]. Similar studies in different parts of Africa have also revealed that wild animals pose major threats to crops and livestock [37–39].

A significant difference in attitudes toward rock hyraxes was observed, correlating with respondents' educational levels. Respondents with higher formal educational backgrounds exhibited more positive perceptions of rock hyrax presence in their area compared to those with no formal education or informal religious schooling. Additionally, male respondents held more positive attitudes toward rock hyraxes than female respondents. Such disparities may reflect varying exposure to ecological concepts or cultural norms. Negative attitudes among local farmers are often influenced by crop damage and livestock predation caused by wildlife, which can undermine support for conservation efforts [38–41].

Most respondents with no formal education or only primary schooling opposed rock hyrax conservation, suggesting limited awareness or concern for protecting the species among these groups. This pattern aligns with concerns raised by landowners in South Africa's Magaliesberg province (North West Region), who similarly view the species negatively [42]. In contrast, a majority of respondents with at least a secondary education supported rock hyrax protection. These findings underscore the pivotal role of formal education in fostering positive attitudes toward wildlife conservation initiatives in the region.

## Conclusions

This study provides crucial information on the population size and habitat association of rock hyraxes in the Zegie Peninsula, as well as the local people's attitude toward rock hyraxes and their ethnozoological importance. We identified the peninsula as an important site for the conservation of the rock hyrax. However, human disturbances, particularly insufficient shelter, affect the population density of the rock hyrax in the area.

Even though rock hyraxes damaged crops, local people use them as traditional medicine to treat human and livestock aliments. This practice plays a vital role for promoting conservation measures for the species. The local administration, along with governmental and non-governmental organizations, should work cooperatively to raise awareness and educate the local community about the economic and ecological importance of rock hyraxes and the need for their conservation. We recommend that local communities cultivate cash crops such as coffee, mangoes, and avocados to mitigate human-wildlife conflicts with rock hyraxes related to crop damage. Local people should create alternative job opportunities, such as fish pond farming and livestock fattening to enhance their economic wellbeing. In addition, it is crucial to raise awareness of the medicinal properties of rock hyraxes among the local community and other stakeholders. Traditional knowledge regarding the therapeutic uses of rock hyrax-derived substances should be documented and scientifically evaluated to validate their efficacy and safety. Engaging local healers, researchers, and healthcare practitioners in collaborative studies can help bridge the gap between indigenous practices and modern medicine. Furthermore, sustainable harvesting practices should be promoted to ensure that the medicinal use of rock hyraxes does not negatively impact their populations. Public education campaigns, workshops, and policy initiatives can facilitate the ethical and sustainable integration of these medicinal resources into local healthcare systems while fostering conservation efforts. These actions serve as habitat management strategies to minimize further alternations and destruction of the rock hyrax habitat. These strategies also help to balance the needs of both the animals and the people. In addition, the study provides baseline information for further research on the medicinal value of rock hyrax, as well as its ecological and behavioral aspects in the region.

## Supporting information

**S1 File. Supporting information.**
(DOCX)

## Acknowledgments

We would like to thank the Zegie district administers for allowing us to carry out this study in the Zegie Peninsula. Our thanks also go to the local people who answered the questionnaire survey politely.

## Author contributions

**Conceptualization:** Zewdu Kifle, Birkie Alehegn.

**Data curation:** Zewdu Kifle, Birkie Alehegn.

**Formal analysis:** Birkie Alehegn.

**Funding acquisition:** Birkie Alehegn.

**Investigation:** Birkie Alehegn.

**Methodology:** Zewdu Kifle, Birkie Alehegn.

**Project administration:** Zewdu Kifle, Birkie Alehegn.

**Resources:** Birkie Alehegn.

**Software:** Birkie Alehegn.

**Supervision:** Zewdu Kifle.

**Validation:** Zewdu Kifle, Birkie Alehegn.

**Visualization:** Birkie Alehegn.

**Writing – original draft:** Birkie Alehegn.

**Writing – review & editing:** Zewdu Kifle.

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
