## [Decision Letter · Decision Letter 0]

15 Oct 2024

Dear Dr. Kifle,

Thank you for submitting your manuscript to PLOS ONE. After careful consideration, we feel that it has merit but does not fully meet PLOS ONE’s publication criteria as it currently stands. Therefore, we invite you to submit a revised version of the manuscript that addresses the points raised during the review process.

We look forward to receiving your revised manuscript.

Kind regards,

Abdallah M. Samy, PhD

Academic Editor

PLOS ONE

Journal Requirements:

1. When submitting your revision, we need you to address these additional requirements. Please ensure that your manuscript meets PLOS ONE's style requirements, including those for file naming. The PLOS ONE style templates can be found at https://journals.plos.org/plosone/s/file?id=wjVg/PLOSOne_formatting_sample_main_body.pdf and https://journals.plos.org/plosone/s/file?id=ba62/PLOSOne_formatting_sample_title_authors_affiliations.pdf 2. Thank you for stating the following financial disclosure: "Bahir Dar University" Please state what role the funders took in the study.  If the funders had no role, please state: "The funders had no role in study design, data collection and analysis, decision to publish, or preparation of the manuscript." If this statement is not correct you must amend it as needed. Please include this amended Role of Funder statement in your cover letter; we will change the online submission form on your behalf. 3. Thank you for stating the following in the Acknowledgments Section of your manuscript: "This work was funded by the Bahir Dar University. We would like to thank the Zegie district administers for allowing us to carry out this study in the Zegie Peninsula. Our thanks also go to the local people who answered the questionnaire survey politely." We note that you have provided funding information that is not currently declared in your Funding Statement. However, funding information should not appear in the Acknowledgments section or other areas of your manuscript. We will only publish funding information present in the Funding Statement section of the online submission form. Please remove any funding-related text from the manuscript and let us know how you would like to update your Funding Statement. Currently, your Funding Statement reads as follows: "Bahir Dar University" Please include your amended statements within your cover letter; we will change the online submission form on your behalf. 4. When completing the data availability statement of the submission form, you indicated that you will make your data available on acceptance. We strongly recommend all authors decide on a data sharing plan before acceptance, as the process can be lengthy and hold up publication timelines. Please note that, though access restrictions are acceptable now, your entire data will need to be made freely accessible if your manuscript is accepted for publication. This policy applies to all data except where public deposition would breach compliance with the protocol approved by your research ethics board. If you are unable to adhere to our open data policy, please kindly revise your statement to explain your reasoning and we will seek the editor's input on an exemption. Please be assured that, once you have provided your new statement, the assessment of your exemption will not hold up the peer review process. 5. We note that Figure 1 in your submission contain [map/satellite] images which may be copyrighted. All PLOS content is published under the Creative Commons Attribution License (CC BY 4.0), which means that the manuscript, images, and Supporting Information files will be freely available online, and any third party is permitted to access, download, copy, distribute, and use these materials in any way, even commercially, with proper attribution. For these reasons, we cannot publish previously copyrighted maps or satellite images created using proprietary data, such as Google software (Google Maps, Street View, and Earth). For more information, see our copyright guidelines: http://journals.plos.org/plosone/s/licenses-and-copyright. We require you to either (1) present written permission from the copyright holder to publish these figures specifically under the CC BY 4.0 license, or (2) remove the figures from your submission: 1. You may seek permission from the original copyright holder of Figure 1 to publish the content specifically under the CC BY 4.0 license.   We recommend that you contact the original copyright holder with the Content Permission Form (http://journals.plos.org/plosone/s/file?id=7c09/content-permission-form.pdf) and the following text:“I request permission for the open-access journal PLOS ONE to publish XXX under the Creative Commons Attribution License (CCAL) CC BY 4.0 (http://creativecommons.org/licenses/by/4.0/). Please be aware that this license allows unrestricted use and distribution, even commercially, by third parties. Please reply and provide explicit written permission to publish XXX under a CC BY license and complete the attached form.” Please upload the completed Content Permission Form or other proof of granted permissions as an ""Other"" file with your submission. In the figure caption of the copyrighted figure, please include the following text: “Reprinted from [ref] under a CC BY license, with permission from [name of publisher], original copyright [original copyright year].” 2. If you are unable to obtain permission from the original copyright holder to publish these figures under the CC BY 4.0 license or if the copyright holder’s requirements are incompatible with the CC BY 4.0 license, please either i) remove the figure or ii) supply a replacement figure that complies with the CC BY 4.0 license. Please check copyright information on all replacement figures and update the figure caption with source information. If applicable, please specify in the figure caption text when a figure is similar but not identical to the original image and is therefore for illustrative purposes only.The following resources for replacing copyrighted map figures may be helpful: USGS National Map Viewer (public domain): http://viewer.nationalmap.gov/viewer/The Gateway to Astronaut Photography of Earth (public domain): http://eol.jsc.nasa.gov/sseop/clickmap/Maps at the CIA (public domain): https://www.cia.gov/library/publications/the-world-factbook/index.html and https://www.cia.gov/library/publications/cia-maps-publications/index.htmlNASA Earth Observatory (public domain): http://earthobservatory.nasa.gov/Landsat:
http://landsat.visibleearth.nasa.gov/USGS EROS (Earth Resources Observatory and Science (EROS) Center) (public domain): http://eros.usgs.gov/#Natural Earth (public domain): http://www.naturalearthdata.com/

Reviewers' comments:

Reviewer's Responses to Questions

**Comments to the Author**

1. Is the manuscript technically sound, and do the data support the conclusions?

Reviewer #1: Partly

Reviewer #2: Yes

Reviewer #3: Yes

2. Has the statistical analysis been performed appropriately and rigorously?

Reviewer #1: No

Reviewer #2: Yes

Reviewer #3: No

3. Have the authors made all data underlying the findings in their manuscript fully available?

Reviewer #1: Yes

Reviewer #2: No

Reviewer #3: Yes

4. Is the manuscript presented in an intelligible fashion and written in standard English?

Reviewer #1: Yes

Reviewer #2: Yes

Reviewer #3: Yes

Reviewer #1: After reviewing the manuscript titled “Population size, habitat association, and local residents’ attitudes towards rock hyrax (Procavia capensis) in Zegie Peninsula, Ethiopia,” I have several recommendations for major revisions:

Key Points and Recommendations:

1. Population estimation methodology

The manuscript uses the line transect method to estimate the rock hyrax population, which is a solid choice. However, it would be helpful to explain a bit more about why 18 transects were used and how their lengths were decided. This would make the study easier to replicate. It’s also great that the transects were done during both the dry and wet seasons and at different times of the day, but it’s not clear whether the same transects were repeated or if new ones were set up each time. Adding some clarification on that, and why those specific times were chosen (maybe based on rock hyrax activity), would really strengthen this section. Finally, including error margins or confidence intervals for the population estimates would boost confidence in the findings and give readers a better understanding of how precise the results are.

2. Data on habitat association

The study gives a general overview of where the rock hyraxes were found, mentioning four habitat types: lakeshore, human premises, tall trees with coffee plantations, and bushland. Most of the hyraxes were spotted around the lakeshore, with none recorded in the bushland. While this gives us a basic idea of their habitat use, the analysis feels a bit surface-level. It would be helpful to go deeper into why they prefer certain areas, like the availability of food or shelter, and back this up with a more detailed analysis. For example, using a habitat preference index or another statistical tool could help show exactly how hyraxes are choosing their habitats. This would make the study more insightful and give a clearer picture of how they’re using different areas throughout the seasons.

3. Questionnaire design and analysis

While the questionnaire survey looks at local residents' attitudes, there isn't much detail on how the questions were developed or validated, and I’m not sure if any pilot testing was done to ensure validity and reliability. Additionally, the analysis leans heavily on descriptive statistics and a simple Chi-square test to compare attitudes across groups like gender and education. Exploring the data further by using more advanced methods, such as logistic regression, could uncover interesting patterns or relationships between other demographic factors and attitudes. This would add depth to the analysis and provide a richer understanding of the local community's views.

4. Lack of ecological context and comparison

The study does mention rock hyrax population densities in other places like the Bale Mountains and the Serengeti, but it doesn’t really dive into the ecological factors that might explain why those numbers are different from what was found in Zegie. Things like food availability, predators, or human activity could be playing a big role, but they aren't discussed in detail. Adding more about these factors would help paint a clearer picture of what’s going on in Zegie. Also, while there are some comparisons with other areas, it would be useful to compare Zegie to regions with similar environments to help explain why the results might be unique. This kind of deeper context would make the findings stronger and more insightful.

5. Cultural considerations

The paper brings up some really interesting cultural beliefs about the rock hyrax, like the idea that it has magical powers and the fact that it’s used in traditional medicine. But this section could be developed more. It would be great to see a deeper discussion of how these beliefs affect people’s attitudes toward conservation. For example, do these beliefs make people less likely to support conservation, or could the use of the hyrax in traditional medicine actually help conservation efforts by showing its value to the community? Exploring this more would give a fuller picture of the cultural dynamics and how they might play a role in protecting the species.

6. Conservation recommendations

The study points out that conservation is important, but the recommendations feel a bit general. It mentions raising awareness and educating the community, which is good, but there could be more specific, practical suggestions. For example, it would be great to see ideas for reducing conflicts between locals and rock hyraxes, like ways to protect crops or involve the community in conservation efforts. You could also suggest some habitat management strategies to help balance the needs of both the animals and the people. Adding more concrete, actionable steps would make the recommendations stronger and more useful for anyone looking to implement conservation measures.

Reviewer #2: General comments

1. The manuscript has availed useful ecologic information concerning Procavia capensis in a peninsula at Lake Bahir Dar shore. It is largely a well written manuscript, but quite a bit of writing problems need attention.

2. Reference citations are a bit awkward especially when a statement begins with a citation number. The name of the author(s) should precede the number reference. For example, the sentence in lines 252/253 state “Similarly, [13] recorded more adult ratio than babies in the Bale Mountains National Park, Ethiopia” Should read as “Teklehaimanot et al (13) recorded more adults than babies in the Bale Mountains National Park, Ethiopia. There are many more examples, e.g., in lines 256, 269, 270, 281, 284, 285, 287, 298, 310, etc.

3. The scientific name for “Shembeko” should be availed to be consistent and to assist international readers grasp the message.

4. The population of rock hyraxes was estimated using 18 strip transects. What is the probability that hyraxes maybe found in different transects at different times and then double-counted? I think a note about possibility (likelihood) of double counting would help. Double counting can lead to overestimation of population size. Contrarily, as the investigators walked along the transect lines, the did not pause at any point. They just kept moving forward. That means resting animals could not be counted. Some behavioral characteristics of hyraxes would determine mobility, e.g., not foraging as a colony, etc.

5. The hours of the hyrax counting seem confusing. How can late afternoon be designated as 8:30 – 11:30? Similarly, 3:30 – 5:30 designated as morning hours need be explicated. If counting started at 3:30 a.m., one wouldn’t expect hyraxes at that early hour. Either a 24-hr or a.m./p.m. time should be provided properly.

6. It would be interesting to see data about the mean number of hyraxes spotted at each counting point.

7. In table 1, it is shown that a high proportion (44/45%) of rock hyraxes dwelled around human dwellings. This observation was not discussed in the results/discussions. It would help to highlight whether humans have intruded in hyrax habitats or if hyraxes liked roaming and foraging around human dwellings.

8. In tables 2 &3 religion is listed under educational level. It is unclear why this is the case. Is it in reference to study participants who can read and write but have not gone to school? This is also the case in figure 2. In both tables and in figure 2, it is better to provide a proper designation of educational level. There is also another designation (certificate and above) that is confusing. It is also important to mention whether schooling level is being attended or completed.

======

For editorial/minor comments, please see attached file, which also contains the above general comments.

Thanks

Reviewer #3: The manuscript brings nice information from under researcher region. But, the concerns raised in the annotated pdf should be addressed line by line. Please see the comments line by line, my comment goes upto page 43

**Do you want your identity to be public for this peer review?** For information about this choice, including consent withdrawal, please see our Privacy Policy

Reviewer #1: No

Reviewer #2: No

Reviewer #3: No

---

## [Author Response · Author response to Decision Letter 1]

21 Nov 2024

09 November 2024

Ethiopia

Ref.:

PONE-D-24-32415

Dear Editor in Chief

PLOS ONE

This is a revision of our manuscript entitled “Population size, habitat association and local residents’ attitude towards rock hyrax (Procavia capensis) in Zegie Peninsula, Ethiopia" by Birkie Alehegn and Zewdu Kifle. On 15 October 2024, we received a major revision request to be acceptable for publication. You invited us to resubmit the revised version of the manuscript that addressed the points raised during the review process.

Hence, we addressed all of the reviewers’ comments within this resubmitted manuscript.

Below there are lists of the revisions.

Sincerely,

Zewdu Kifle,

corresponding author

Response for Editor and reviewers

Thank you for your critical comments. I highlighted revisions with green color. In addition, I highlighted with red color and strikethrough those deleted letters, words, phrases, or sentences. In addition, I provided reviewers’ concerns with justification within this file.

Thank you!

Response for Editor

Thank you! I have revised in detail on the presentation of this manuscript.

Response for reviewers

Reviewer #1:

Thank you for your important comments.

1. Population estimation methodology

Thank you! We determined the lengths and the numbers of transects in proportion to the size of the habitat strata. We repeated the same transects at each time of the day and season. We clarified why we have chosen those specific times of the day as the rock hyrax is mostly active for foraging and sun basking at these times of the day. We stated this in the revised version of the manuscript. We included confidence intervals.

2. Data on habitat association

Thank you! We noted why we did not record rock hyraxes in the bushland habitat. This habitat might not be suitable for foraging and hiding sites like rock outcrops. However, a habitat preference index or another statistical tool for the animal requires more habitat characteristics data. This parameter needs additional research to answer the question in more detail. Thus, we recommended to do further ecological research in the area

3. Questionnaire design and analysis

Thank you again! We performed a preliminary survey to ensure validity and reliability in the area. This pilot test was checked through data collection from 17 respondents. We stated the information in this revised version of the manuscript. We analyzed the data using the Chi-square test of independence to relate educational level and other factors to attitudes

4. Lack of ecological context and comparison

Food availability, better hiding places, and high human disturbances are considered factors for the low density of rock hyrax in the present study area compared to protected areas like the Bale Mountain National Park in Ethiopia and Serengeti National Park in Tanzania. We have stated such ecological aspects in the discussion section.

5. Cultural considerations

Thank you! It is very a fascinating cultural belief of the local community about the rock hyrax. That is why about 42.4% of the respondents had a good attitude towards rock hyrax in the study area. However, still more respondents had negative attitudes toward the animal. Yes, you are right. The traditional medicine value of the rock hyrax helps to promote conservation efforts in the region.

6. Conservation recommendations

Thank you! We added actionable recommendations to balance the needs of both the animals and the local people in the area

Reviewer #2

General comments:

Thank you for your critical comments on this research work.

1. Thank you for appreciating the usefulness of the study.

2. Thank you! We corrected all reference citations.

3. Thank you we deleted this local name of rock hyrax ‘Shembeko’

4. Thank you here again! To avoid double counting, we counted consecutive transects simultaneously. The distance between two consecutive transect lines was 200 to 300 m to avoid double counting. Therefore the likelihood of double counting is almost zero. We scanned the habitat of rock hyrax within the strip with brief pauses of two minutes every 50 m. We stated this in the revised version of the manuscript.

5. Thank you! Sorry, we used the Ethiopian time zone. We corrected it.

6. We put the overall mean number of rock hyraxes of each habitat stratum.

7. Thank you! We added the reason why we counted a high proportion (44/45%) of rock hyraxes in the human dwellings in the discussion section.

8. In tables 2 &3 religion, these are deacons and priests who learned religion through the Ethiopian Orthodox Church. We designed respondents who completed grade 12 as certificates and above.

Reviewer #3:

Thank you for appreciating the manuscript’s information. I corrected concerns that you raised in the annotated pdf.

Specific comments:

1. Line 11: Thank you! We changed the word ‘well’ with ‘in detail’.

2. Line 27 and 33. We changed the word ‘region’ into area

3. Line 101 and 102: Thank toy! We rewrote the sentences.

4. Line 16: We deleted ‘Shenbeko’.

5. Line 168: We deleted ‘a total mean of’ and we changed ‘Zegie Peninsula’ with study ‘area’.

6. Line 191: Thank you! We added ‘a’ before ‘few’.

7. Line 195-201: We edited based on your comment.

8. Line 204-205: We corrected

9. Line 219: We deleted ‘some’ from the sentence.

10. Line 229-220: We edited the sentence.

11. Line 247: Thank you! We corrected the sentence based on your comments.

12. Line 246: We edited the phrase ‘habitat disturbance’.

13. Line 251-254: we corrected those sentences. We put the reference ‘13’ as the end of the sentence.

14. Lien 256: We put the reference at the end of the sentence.

15. Line 259: We corrected it.

16. Line 262: We deleted ‘in the region’.

17. Line 267: We edited it.

18. Line 269 and 270: We put these references at the end of the sentence.

19. Line 280: We deleted ‘save’

20. Line 282, 284, 285, and 287: We put the reference at the end of the sentence.

21. Line 286: We deleted letter ‘y’ from rocky.

22. Line 292: We edited this sentence.

23. Line 295: Thank you! We edited this sentence.

24. Line 297 and 298: We edited these sentences.

25. Line 298: We put the reference at the end of the sentence.

26. Line 300-301: We deleted this sentence.

27. Line 303: We deleted the word ‘background’.

28. Line 309: We deleted the word ‘enough’.

29. Line 310: We put the reference at the end of the sentence.

30. Line 312: We corrected this sentence.

31. Line 317: Thank you! We corrected this sentence.

32. Line 332: We corrected this sentence.

---

## [Decision Letter · Decision Letter 1]

7 Mar 2025

Dear Dr. Kifle,

Thank you for submitting your manuscript to PLOS ONE. After careful consideration, we feel that it has merit but does not fully meet PLOS ONE’s publication criteria as it currently stands. Therefore, we invite you to submit a revised version of the manuscript that addresses the points raised during the review process.

We look forward to receiving your revised manuscript.

Kind regards,

Honnavalli Nagaraj Kumara, Ph.D.

Academic Editor

PLOS ONE

Additional Editor Comments :

Both the reviewers provided the corrections and suggestions directly in the MS, and the same is attached with this email. 

Reviewers' comments:

Reviewer's Responses to Questions

**Comments to the Author**

Reviewer #2: (No Response)

Reviewer #3: All comments have been addressed

2. Is the manuscript technically sound, and do the data support the conclusions?

Reviewer #2: Yes

Reviewer #3: Partly

3. Has the statistical analysis been performed appropriately and rigorously?

Reviewer #2: Yes

Reviewer #3: Yes

4. Have the authors made all data underlying the findings in their manuscript fully available?

Reviewer #2: No

Reviewer #3: Yes

5. Is the manuscript presented in an intelligible fashion and written in standard English?

Reviewer #2: No

Reviewer #3: Yes

Reviewer #2: The manuscript can still be improved substantially. There are some corrections to be made. I have used the pdf file to directly put my comments at the appropriate locations. I refer the authors to go through the revised document where I have shown my comments and corrections.

Reviewer #3: The MS should address the weakness before being accepted for publication. I have attached a more detailed list of comments and suggestions within the manuscript itself [if applicable, mention how you've provided comments, e.g., using track changes, PDF annotations]. These comments are intended to be constructive and to help the authors strengthen their work.

I believe that addressing these points will significantly improve the quality and clarity of the manuscript. Therefore, I recommend that the manuscript be revised before being considered further for publication.

**Do you want your identity to be public for this peer review?** For information about this choice, including consent withdrawal, please see our Privacy Policy

Reviewer #2: No

Reviewer #3: No

---

## [Author Response · Author response to Decision Letter 2]

14 Apr 2025

14 April 2025

Ethiopia

Ref.:

PONE-D-24-32415R1

Dear Editor in Chief

PLOS ONE

This document constitutes a revised version of our manuscript titled “Population Size, Habitat Association, and Local Residents’ Attitudes Toward Rock Hyrax (Procavia capensis) in Zegie Peninsula, Ethiopia” by . Birkie Alehegn and Zewdu Kifle. On 7 March 2025, we received a revision request to make the manuscript acceptable for publication. As per your invitation, we have resubmitted the revised manuscript after thoroughly addressing all concerns and suggestions raised during the peer-review process.

Accordingly, we have addressed all reviewers’ comments in this resubmitted manuscript. Below is a summary of the revisions made:

Sincerely,

Zewdu Kifle,

Corresponding author

Response for Editor and reviewers

Thank you for your critical feedback. I have addressed your comments by highlighting revisions in green and marking deleted text (letters, words, phrases, or sentences) in red with strikethrough. Additionally, I have included reviewers’ concerns alongside corresponding justifications within this document. To enhance the manuscript’s quality and clarity, I have also corrected grammatical errors throughout the text.

Thank you!

Response for Editor

Thank you! I have revised in detail on the presentation of this manuscript.

Response for reviewers

Reviewer #1:

Thank you for your important comments.

Line 44: Thank you! I edited the sentence and changed the word ‘harvest’ with ‘crop cultivation’.

Line 62: I added “particularly “

Line 102: Thank you! I deleted this phrase (“the interviewer fully”) when I edited the sentence.

Line 102: I deleted “about”.

Line 103: I deleted “on”.

Line 105: Thank you! I changed “being explained” with “they received a detailed explanation”

Line 114: Thank you for checking the spelling error. I corrected it

Line 121: I changed “are” with “includes”.

Line 144: Thank you! You are right. It is expected that rock hyraxes start their day before 3:00 a.m. However, for this particular research, we collected data starting with this time.

Line 145: Thank you so much!! I edited the time zone with 2:30-5:00 p.m.

Line 175: Thank you! Yes, zero is still the valid observation. However, since it is an extremely outlier value, we excluded it from the analysis.

Line 192-193: Thank you a lot! I corrected this sentence

Line 198-200: Thank you so much again! I corrected the age ratio here too.

Line 224: I edited the phrase.

Line 243: Thank you! I rewrote the sentence.

Line 248: Thank you! I edited the phrase

Line 252: I put this sentence at the end of the paragraph.

Line 259: Thank you! I have edited this word.

Line 261: Thank you! I have edited this word.

Line 262: I changed the word “informed” with “mentioned”.

Line 286: Thank you! I rewrote this sentence.

Line 295: I changed the word “case” with “cause”.

Line 300-301: Thank you! I deleted this sentence from here.

Line 307: I deleted this phrase.

Lie 312: I deleted the phrase “and birth”.

Line 318: I deleted the phrase.

Line 319-320: Thank you! I deleted this phrase.

Line 327: I deleted the word “also”

Line 328: I changed “this might be” with” likely”

Line 335: I added “in the Zegie Peninsula” here.

Line 336: I edited this phrase.

Line 338: Thank you! I deleted the word “develop” and edited it.

Line 342: I preferred the word “behavior” rather as it is

Line 355: I reshuffled this sentence

Line 358: I changed “uneducated” with no formal education

Line 360: Thank you! I edited this sentence.

Line 368: I added “in the Zegie Peninsula” here.

Line 369-370: I deleted this phrase

Line 371: Thank you! I edited this phrase.

Line 381: I preferred the word “fattening” as it is.

Line 381-382: Thank you so much! I rewrote this sentence to emphasis the medical value of rock hyraxes.

Line 383: I deleted “the”.

Line 385: I added “in addition” here.

Line 513: I changed “uneducated” into “no education” with the table.

Line 516: I edited it.

Line 516: I changed “uneducated” into “no education” with the graph.

---

## [Editor Report · Decision Letter 2]

16 Apr 2025

Population size, habitat association and local residents’ attitude towards rock hyrax (Procavia capensis) in Zegie Peninsula, Ethiopia

PONE-D-24-32415R2

Dear Dr. Kifle,

We’re pleased to inform you that your manuscript has been judged scientifically suitable for publication and will be formally accepted for publication once it meets all outstanding technical requirements.

Kind regards,

Honnavalli Nagaraj Kumara, Ph.D.

Academic Editor

PLOS ONE

Additional Editor Comments (optional):

All the concerns are well addressed
---

## [Editor Report · Acceptance letter]

PONE-D-24-32415R2

PLOS ONE

Dear Dr. Kifle,

I'm pleased to inform you that your manuscript has been deemed suitable for publication in PLOS ONE. Congratulations! Your manuscript is now being handed over to our production team.

Kind regards,

on behalf of

Dr. Honnavalli Nagaraj Kumara

Academic Editor

PLOS ONE